# Mitochondrial Pathobiology and Metabolic Remodeling in Progression to Overt Systolic Heart Failure

**DOI:** 10.3390/jcm9113582

**Published:** 2020-11-06

**Authors:** Antoine H. Chaanine, Thierry H. LeJemtel, Patrice Delafontaine

**Affiliations:** 1Department of Medicine/Heart and Vascular Institute, Tulane University, New Orleans, LA 70112, USA; lejemtel@tulane.edu (T.H.L.); pdelafon@tulane.edu (P.D.); 2Department of Physiology, Tulane University, New Orleans, LA 70112, USA; 3Department of Pharmacology, Tulane University, New Orleans, LA 70112, USA

**Keywords:** heart failure, remodeling, mitochondria, calcium, oxidative capacity, metabolism

## Abstract

The mitochondria are mostly abundant in the heart, a beating organ of high- energy demands. Their function extends beyond being a power plant of the cell including redox balance, ion homeostasis and metabolism. They are dynamic organelles that are tethered to neighboring structures, especially the endoplasmic reticulum. Together, they constitute a functional unit implicated in complex physiological and pathophysiological processes. Their topology in the cell, the cardiac myocyte in particular, places them at the hub of signaling and calcium homeostasis, making them master regulators of cell survival or cell death. Perturbations in mitochondrial function play a central role in the pathophysiology of myocardial remodeling and progression of heart failure. In this minireview, we summarize important pathophysiological mechanisms, pertaining to mitochondrial morphology, dynamics and function, which take place in compensated hypertrophy and in progression to overt systolic heart failure. Published work in the last few years has expanded our understanding of these important mechanisms; a key prerequisite to identifying therapeutic strategies targeting mitochondrial dysfunction in heart failure.

## 1. Introduction

The heart is an organ requiring high-energy to meet its metabolic demands and has the plasticity to undergo remodeling, hypertrophy or atrophy, in response to variable stimuli [1]. Mitochondria, which are considered the powerhouse of the cell, are mostly abundant in the heart, and have the densest cristae. Their function extends beyond energy production to include participation in metabolism, signaling and redox balance and ion homeostasis. Thus, they are master regulators of cell survival or cell death [2].

The heart has the flexibility to switch its preference in substrate utilization and metabolism based on its energetic need/demand and substrate availability. Under normal conditions, 60–90% of adenosine triphosphate (ATP), generated by oxidative phosphorylation, comes from fatty acid utilization and β-oxidation and the remaining from glucose metabolism. Ketones are substrates that are utilized in periods of prolonged starvation. Metabolic remodeling, extending from altered mitochondrial dynamics and function to impaired substrate utilization and metabolism, is an important pathophysiological process that plays a role and contributes to myocardial remodeling in heart failure (HF). In this manuscript, we will review and discuss molecular mechanisms involved in pathophysiological processes in mitochondria as they pertain to changes in their morphology and dynamics, oxidative capacity and metabolism that take place in compensated hypertrophy, in transition to and at systolic HF development.

## 2. Mitochondrial Morphology and Dynamics in HF

Mitochondria are dynamic structures that undergo cycles of fission and fusion, which is essential for mitochondrial biogenesis and mitochondrial quality control [3,4]. Damaged mitochondria that undergo fission and are unable to fuse again are eliminated and recycled via mitophagy [5]. The balance between mitochondrial fission and fusion is maintained under normal conditions, and is orchestrated by mitochondrial fusion proteins at the outer mitochondrial membrane (OMM): mitofusin 1 (MFN1) and mitofusin 2 (MFN2) [6], and inner mitochondrial membrane (IMM): optic atrophy 1 (OPA1) [7]; and by mitochondrial fission proteins: dynamin related protein 1 (DRP1) [8] and fission 1 at the OMM. It is not yet clearly understood what promotes mitochondrial fission at the IMM. Figure 1A is a transmission electron photomicrograph of a normal left ventricular (LV) myocardium. Please refer to figure legend for details. In HF, the balance tips towards mitochondrial fission and mitochondrial fragmentation, Figure 1B. Mitochondrial fission and fragmentation is observed as early as week 2, following ascending aortic banding in rat [9]. Mitochondrial fragmentation worsens as the animals transition from a compensated phenotype into a remodeled overt systolic HF phenotype [9], known as HF with reduced ejection fraction (HFrEF). In addition to the decrease in mitochondrial area and mitochondrial cristae density, derangement in mitochondrial distribution is observed. The fragmented intermyofibrillar mitochondria appear to be arranged in clusters within the adjacent sarcomeres, a finding that is observed early on in pathological hypertrophy in transition to HFrEF [10], Figure 1B. Moreover, mitochondria at different stages of vacuolar degeneration are encountered where there is gradual loss of mitochondrial cristae and mitochondrial swelling, eventually leading to rupture of the OMM at a more advanced stage of mitochondrial vacuolar degeneration [11,12], Figure 1B. Stages of mitochondrial vacuolar degeneration, described elsewhere [11], are shown in Figure 2. Similar mitochondrial morphological changes have been described with mitochondrial calcium overload [13,14], and in cells undergoing apoptosis [11,15]. These mitochondrial morphological changes were also observed in human HF and were more severe in patients with HFrEF, compared to patients with HF and preserved ejection fraction (HFpEF) related to ischemic heart disease or severe aortic stenosis [12]. Moreover, within the areas of clustered mitochondria, often numerous lysosomes are observed fusing with the surrounding mitochondria pointing towards active mitophagy [12,14], Figure 1B. Molecular markers of autophagy and mitophagy are upregulated early on in compensated pathological hypertrophy or HFpEF, and intensify with the progression to HFrEF [9,12,16]. At the initial stage of remodeling, such as at compensated hypertrophy, mitophagy may play a cardioprotective role by maintaining mitochondrial quality control. However, with the advancement of cardiac remodeling and activation of signaling pathways that induce mitochondrial dysfunction and apoptosis, the mitophagy system becomes overwhelmingly saturated and insufficient to maintain a healthy mitochondrial population [16,17]. This is supported by the observation that the mitochondria in HFrEF are at a relatively advanced stage of vacuolar degeneration compared to a normal or phenylephrine-stressed adult cardiac myocyte for 2 h, and HFpEF, Figure 3A–C. The functional capacity of the HFrEF patients, from whom the subepicardial biopsy was obtained, were in the category of New York Heart Association (NYHA) class III and IV, compared to HFpEF patients who the majority were classified as NYHA class II or IIb [12]. Moreover, there is dysfunction of the lysosomes due to peroxidation of the lysosomal membranes and accumulation of lipofuscin material within the lysosomal body [18,19,20], which renders the autophago-lysosomal system dysfunctional.

The etiology of mitochondrial clustering, molecular mechanisms driving it and its (patho)-physiological effect on mitochondrial function and inter-organelle tethering and signaling in HF has not been elucidated. Previous work has shown that the clustered mitochondria protein homolog (Cluh), which is an mRNA-binding protein involved in proper cytoplasmic distribution of mitochondria, binds mRNAs of nuclear-encoded mitochondrial proteins in the cytoplasm and regulates transport or translation of these transcripts close to mitochondria, playing a role in mitochondrial biogenesis and oxidative capacity [21]. Mice deficient in Cluh failed to adapt to conditions of nutrient deficiency, such as during starvation or transition from fetal to neonatal period, and died shortly after birth [22]. In these mice, Cluh deficiency was found to affect mitochondrial respiratory function, mainly electron transport chain complex I activity, mitochondrial DNA content and cristae density were decreased in both liver and heart [22]. Similar findings were observed in HeLa cells [23]. Our unpublished proteomic data suggest that Cluh expression is downregulated in a rat HF model of moderate remodeling and early systolic dysfunction (MOD) and in HFrEF. It would be interesting to explore the role of Cluh in the pathogenesis of HF and mitochondrial dysfunction in future studies.

Changes in the expression and post-translational modification (PTM) of mitochondrial dynamic proteins (MDPs) are evident in HF. Previous work suggests that expression of MFN1 and MFN2 is increased in patients with both ischemic and non-ischemic cardiomyopathies [24]. MFN2 expression was unchanged in rat pressure overload (PO)-induced HF model and in LV subepicardial biopsies from patients with HFpEF and HFrEF related to ischemic heart disease (IHD) or severe aortic stenosis (AS) compared to normal [12,14]. Recent work suggested a role for MFN1 PTM in HF. Phosphorylation of MFN1 by beta II protein kinase C (βIIPKC) at serine 86 resulted in partial loss of its GTPase activity and in build-up of fragmented and dysfunctional mitochondria in HF [25]. Inhibition of MFN1-βIIPKC association by a peptide called, SAMβA, restored mitochondrial morphology and function and improved cardiac contractility in a rat model of HF [25]. MFN2 knockout has been shown to be cardioprotective in the acute phase of ischemia reperfusion injury by attenuating MFN2-mediated tethering of the sarcoplasmic reticulum to neighboring mitochondria, therefore reducing mitochondrial calcium overload in a MFN1 and MFN2 double knockout (DKO) mouse model [26]; although the mitochondria were fragmented with decreased mitochondrial oxidative capacity and impaired myocardial contractile function in the MFN DKO mice, compared to WT mice [26]. Moreover, stress induced phosphorylation of MFN2 at serine 27 by the mitogen-activated protein kinase, JNK, promoted MFN2 proteasomal degradation and facilitated mitochondrial fragmentation and apoptosis [27]. OPA1, beside its role in fusion of the IMM, has been shown to play a protective role by maintaining mitochondrial cristae integrity and morphology; thus, playing a role in mitochondrial respiratory efficiency [28]. It has been shown that OPA1 expression was decreased or unchanged in human HFrEF [12,24]. Additionally, it has been shown that the mitochondrial death and mitophagy marker, BNIP3, binds and inhibits OPA1 via its transmembrane domain promoting mitochondrial fragmentation and apoptosis [29]. Moreover, BNIP3 knockdown in phenylephrine stressed cardiac myocytes enhanced DRP1 phosphorylation at serine 637, therefore promoting its cytoplasmic translocation [14]. DRP1 expression was increased in human HFrEF [12,24]. These data suggest that MDPs are tightly regulated by a series of signaling pathways that regulate their transcription, PTM and proteasome-dependent degradation; therefore, modulating mitochondrial morphology, dynamics and function [30]. It remains unclear which of the MDPs is a major player in mitochondrial dysfunction and whether altering MDPs expression versus their PTM would be a better therapeutic strategy in HF. For instance the mitogen activated protein kinase, JNK, which is activated under many conditions of cardiac stress [9,31,32], phosphorylates MFN2 at S27 promoting its degradation, and is a key player in myocardial remodeling by contributing to endoplasmic reticulum (ER) stress and mitochondrial induced apoptosis [9,33,34]. Similarly, protein kinase A (PKA) and Calcineurin have opposing effects, and share many target proteins, such as DRP1 [35,36], Phospholamban (calcium cycling), Troponin I (contractile protein) and glycogen synthase, and are key players in the initiation and propagation of cardiac remodeling and progression to HF [37,38].

## 3. Mitochondrial Matrix Calcium, Redox Balance and Oxidative Capacity in HF

Mitochondrial oxidative phosphorylation is a coupled process between the tricarboxylic acid (TCA) cycle and the electron transport chain (ETC) system. Byproducts of this process are the generation of adenosine triphosphate (ATP), carbon dioxide and reactive oxygen species (ROS). The majority of ROS are produced by the ETC complexes I and III in the form of oxygen radicals, which are reduced to hydrogen peroxide (H_2_O_2_) by the mitochondrial manganese superoxide dismutase (SOD2). H_2_O_2_ is then reduced to water by the mitochondrial or cytosolic antioxidant systems, such as catalase, thioredoxin-peroxidase and glutathione-peroxidase. Calcium is a key second messenger that orchestrates the interplay of mitochondrial redox and energetics with the excitation contraction coupling [39]. Increases in cytosolic and mitochondrial calcium transients have been observed upon acute stimulation of the cardiomyocyte with isoproterenol and increases in energetic demand. This serves as a stimulus to activate enzymes in the TCA cycle and ETC complexes, such as pyruvate dehydrogenase, NAD^+^-dependent isocitrate dehydrogenase, α-ketoglutarate dehydrogenase and ETC complexes I and III [40], respectively. Therefore, the same signal that stimulates muscle contraction, also stimulates ATP production to meet the higher energetic needs/demands, but at the expense of increased ROS generation. Crosstalk exists between calcium and ROS signaling systems. Together, they can regulate and fine tune cellular signaling networks [41]. Derangements in either of the aforementioned signaling systems would adversely affect the other, and will have detrimental consequences on cellular fate and survival [41], as we discuss below.

Mitochondrial calcium uptake is via the voltage dependent anion channel, VDAC1, at the OMM, and is tightly regulated by the mitochondrial calcium uniporter (MCU) at the IMM [42,43]. Mitochondrial calcium efflux is via the sodium/calcium/Lithium exchanger (NCLX) at the IMM into the intermembranous space and then through VDAC1 into the cytoplasm [44]. Mitochondrial matrix calcium overload and increased ROS are now being realized as important players in mitochondrial dysfunction and decreased oxidative capacity in HF [44]; despite the paradox of decrease in MCU level and mitochondrial calcium uptake and oxidative capacity in diabetic cardiomyopathy [45]. Additionally, our unpublished proteomic data suggest a decrease in MCU and its subcomplex mitochondrial calcium uptake 1 (Micu1) expression in HFrEF, but not at earlier stages of cardiac remodeling, where an increase in mitochondrial-matrix calcium has been observed [10]. Could this paradox be explained by activation of signaling pathways leading to modulation and/or PTM of mitochondrial uptake proteins at the OMM and IMM? This area may need to be elucidated in future studies. It has been shown that recombinant expression of VDAC enhanced ER-mitochondrial contact sites and calcium transfer into the mitochondria and apoptosis [46,47]. In HF, VDAC expression is not changed, however, VDAC oligomerization induced by the Bcl2 like protein, BNIP3 [10], and in response to apoptotic stimuli [48], possibly through its PTM, may alter calcium entry into the intermembranous space and subsequently into the matrix leading to mitochondrial matrix calcium overload, mitochondrial dysfunction and apoptosis. Moreover, previous work has shown that calcium uptake via MCU is enhanced by its PTM by the mitochondrial-CaMKII [49,50]. Whether other modes of calcium entry at the IMM exists through a different channel other than the MCU, needs to be elucidated in the future. Increase in mitochondrial ROS is the result of mitochondrial calcium overload-induced ROS production and decrease in the ROS scavenging system, due to the decrease in SOD2 expression and activity in HF [51].

Previous work suggested that mitochondrial biogenesis and oxidative capacity were preserved in patients with HFpEF [12], or enhanced in pathological compensated hypertrophy [52,53]. There is a decline in mitochondrial content and oxidative capacity in rodent models upon transitioning from a compensated state to moderate remodeling and early systolic dysfunction [53]. Cytochrome C oxidase activity is decreased in early systolic dysfunction, along with a decrease in expression of the ETC complexes I and IV [14]. Transitioning to HFrEF is marked by a decrease in mitochondrial biogenesis and PGC-1α expression, along with decrease in expression of the ETC complexes. These findings were observed in an animal models and in human HFrEF [12,52]. Another mechanism of decreased oxidative capacity in HF is PTM of mitochondrial proteins, mainly hyperacetylation. Increased mitochondrial protein acetylation has been observed in animal models and in human HFrEF [54,55] affecting proteins related to fatty acid beta-oxidation, TCA cycle and ETC complexes leading to decrease in their activity. The etiology of mitochondrial protein hyperacetylation in HF is poorly understood and is speculated to be related to the presence of excessive acyl-CoAs and reduced protein deacetylation by the sirtuin family of NAD^+^-dependent deacetylases, mainly SIRT3 [56], which is the predominant deacetylase, and SIRT5. Our unpublished shotgun proteomic data support above findings and suggest that both, SIRT3 and SIRT5, were downregulated in rat HFrEF phenotype, but not in pathological compensated hypertrophy or in early systolic dysfunction and moderate remodeling.

## 4. Metabolic Remodeling in HF

Metabolic remodeling take place early on in pathological hypertrophy and its profile differs with the progression to an advanced HF phenotype [57]. This is largely related to the change in expression levels of metabolic enzymes and their PTM. It has been shown that fatty acid uptake and fatty acid oxidation (FAO) are adversely affected in pathological hypertrophy in transition to HF, with a noticeable increase and reliance on glucose uptake and glucose metabolism at least in the initial stages of HF [58,59]. This is related to suppression of genes, such as PPAR-α and PGC-1α, regulating the expression of important FAO proteins [60,61]. Moreover, the activity of PGC-1α is highly regulated by both its expression levels and PTM such as phosphorylation, acetylation or methylation [62]. As HF progresses to an advanced stage, there is downregulation of genes related to glucose metabolism as well [57,63,64,65]. This metabolic remodeling at the initial stages of HF could be adaptive as ATP generation from glycolysis and glucose metabolism is more efficient in terms of oxygen consumption compared to fatty acid metabolism [51]. However, the amount of ATP production from glucose metabolism is limited, and may not be sufficient to meet the metabolic demand and increase in wall stress. Moreover, impaired fatty acid metabolism and beta-oxidation leads to the accumulation of harmful lipid products in the cytoplasm such as ceramide and is associated with cardiolipotoxicity [66]. Our unpublished metabolomics data suggest that mitochondrial fatty acid β-oxidation is impaired in a rat MOD HF phenotype with a substantial increase in and reliance on glucose metabolism. Both, fatty acid β-oxidation and glucose metabolism, are impacted upon transitioning to HFrEF. Recently, it has been observed that ketone body oxidation was increased in both animal models and human HF [51,65,67,68]. The increase in ketone uptake and metabolism in HF is related to elevated circulating ketone levels in patients with HF, suggesting a change in systemic metabolism. Published work suggests that increased ketone body oxidation is cardioprotective, however, the underlying mechanism is not yet clear and need to be further elucidated. Moreover, it has been shown that reactome pathways related to branched chain amino acid (BCAA) catabolism are downregulated in compensated hypertrophy and contributes to the transition to HFrEF [69,70]. The accumulation of BCAAs also impairs glucose metabolism and worsens ischemia/reperfusion injury, and is rescued by restoring BCAAs catabolism [71]. Less is known about the importance of altered amino acid (AA) [65] and nucleotide metabolism [72] in the contribution and development of HF. Given that AA and nucleotides contribute less to ATP production, but are involved in other cellular processes, such as signaling, redox balance and calcium homeostasis [73], it would be intriguing to further characterize the changes in metabolic profile of these compounds at different stages of HF development and unravel the molecular mechanisms and processes involved. This is supported by previous studies showing that AA [74,75] and BCAA [76] supplementation paradoxically improved functional capacity in HF patients, despite the abundance of these metabolites in the serum of patients with HF [65]. Furthermore, given that a majority of proteins involved in metabolism, which are downregulated in HF, are mitochondrial proteins and given the enhanced mitochondrial vacuolar degeneration and degradation in HF, it would be interesting to explore whether therapies targeting mitochondrial dysfunction and degradation in HF would favorably affect and restore, to a certain extent, some of the myocardial metabolic deficiencies encountered during the progression and development of HFrEF.

There is a growing line of advance that cardiac mitochondria are implicated in the pathogenesis of cardiac arrhythmias [77,78,79,80,81], beyond their role in the pathogenesis of HF and mitochondrial cardiomyopathies [82]. Moreover, skeletal muscle mitochondrial pathology is well recognized to play a fundamental role in the decline in functional capacity of HFpEF patient population and decreased peak oxygen consumption during exercise [83,84]. This is due to the heightened oxidative stress conditions associated with coexisting comorbidities in the HFpEF patient population, such as hypertension, diabetes and obesity [85]. Contrary to HFpEF, HFrEF patient population suffer from skeletal muscle mitochondrial dysfunction and muscle wasting, known as cardiac cachexia, which has been associated with poor outcomes and decreased survival [86], and is likely related to failure to thrive, due to poor cardiac output and neurohormonal disturbances. The aforementioned conditions are out of the scope of this paper and will not be discussed further.

## 5. Conclusions

Mechanisms leading to impaired mitochondrial function in HF are multiple and complex. They include a spectrum of pathophysiological processes that synergistically affect mitochondrial function. These pathophysiological processes take place simultaneously or subsequently at different stages of HF, worsen with and contribute to the progression to overt systolic HF phenotype, Figure 4 and Figure 5. Significant work has been carried out thus far, which has expanded our knowledge and understanding of mitochondrial pathobiology in HF. However, there remain unanswered questions, as highlighted above, that have not been clearly elucidated and warrant further investigation. Further understanding of these mechanisms will aid in developing best therapeutic strategies to target and treat mitochondrial dysfunction in HF.

## Figures and Tables

**Figure 1 jcm-09-03582-f001:**
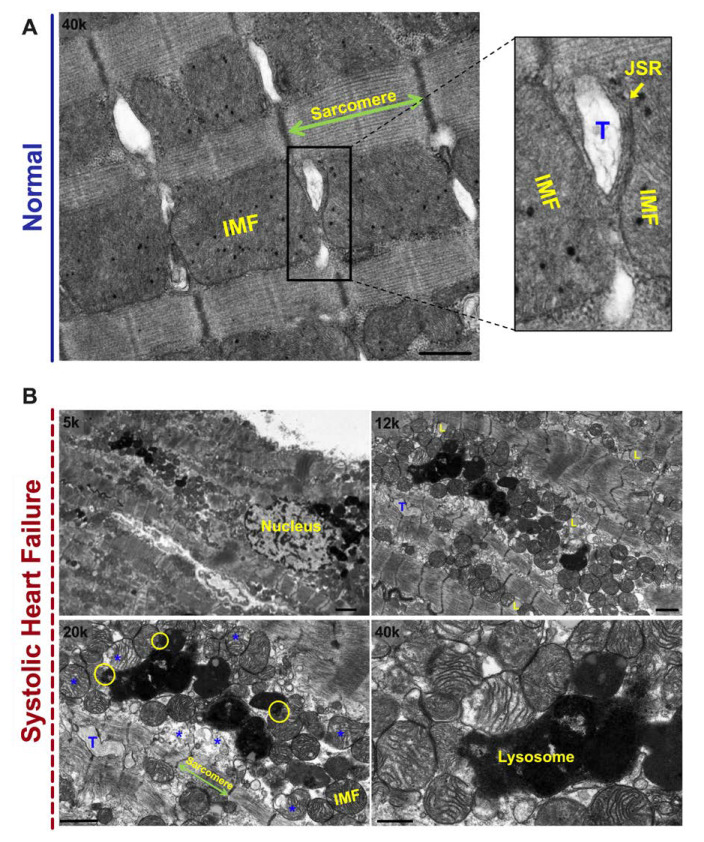
Mitochondrial ultrastructural changes in heart failure. (**A**) Transmission electron micrograph of longitudinal section of left ventricular (LV) myocardium suggests that each intermyofibrillar mitochondrion extends from one Z band to another, and neighbors with adjacent sarcomeres. The T-tubules (T), which are invaginations of the plasma membrane, are surrounded by the junctional sarcoplasmic reticulum (JSR), and are interspersed between adjacent intermyofibrillar mitochondria. The longitudinal sarco/endoplasmic reticulum runs in a network configuration between adjacent intermyofibrillar mitochondria and sarcomeres. This topology of the intermyofibrillar mitochondria places them at the hub of calcium signaling by being physically tethered and functionally coupled with the adjacent organelles, particularly the endoplasmic reticulum. Photomicrograph is 40k X magnified, scale bar 0.5 μm. (**B**) Transmission electron micrograph of LV myocardium in systolic HF (HFrEF). Please refer to text in the body of manuscript for details. Yellow circles showing areas where lysosomes are fusing with surrounding mitochondria. Blue asterisks showing mitochondria at different stages of vacuolar degeneration. Additionally, there is accumulation of lipid (L) particles in the cytoplasm in heart failure (HF) myocardium. Very few lipid particles are seen in normal myocardium. Images are 5k, 12k, 20k and 40k X magnified, scale bar 2 μm, 1 μm, 1 μm and 0.5 μm, respectively. IMF: intermyofibrillar mitochondria, JSR: junctional sarcoplasmic reticulum, T: T-tubule. Figure 1A is adapted and modified with permission from Stem Cell and Gene Therapy for Cardiovascular Disease (2016), Chapter 30, Chaanine et al.

**Figure 2 jcm-09-03582-f002:**
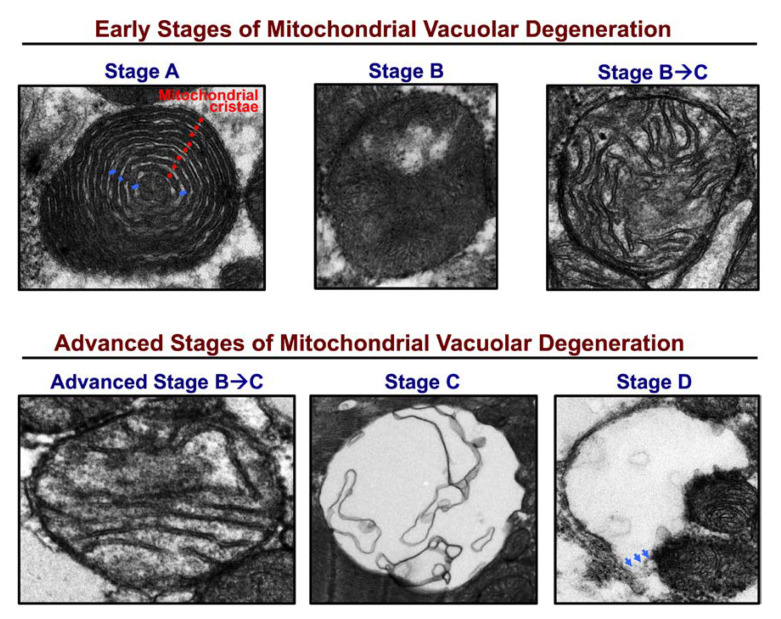
Stages of mitochondrial vacuolar degeneration. (**Stage A**) Concentric remodeling of mitochondrial cristae (red dots) in an onion like pattern with widening of spaces between them (blue dashes). **Stage A** is encountered in stressed adult cardiac myocytes in vitro and is not appreciated in animal models of HF in vivo. (**Stage B**) Loss of mitochondrial cristae in a single mitochondrion area. (**Stage B ➔ C**) Loss of mitochondrial cristae in multiple mitochondrion areas. The inner mitochondrial membrane (IMM) and outer mitochondrial membrane (OMM) remain intact and there is no evidence of mitochondrial swelling. Advanced (**Stage B ➔ C**) mitochondrial swelling with advanced loss of mitochondrial cristae in multiple mitochondrion areas and permeabilization of IMM. (**Stage C**) Mitochondrial swelling and complete dissolution of mitochondrial cristae and IMM. (**Stage D**) include morphological changes in **Stage C**, along with rupture of the OMM (blue arrows). Image obtained and modified in permission from Chaanine et al. [11].

**Figure 3 jcm-09-03582-f003:**
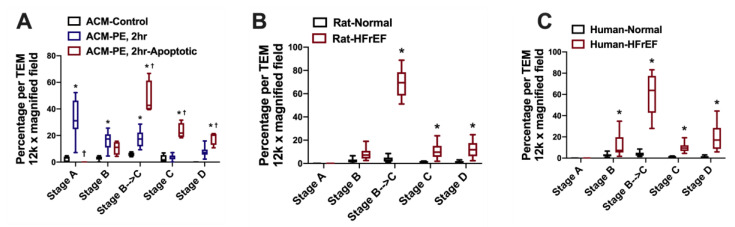
Percentage of mitochondria at different stages of mitochondrial vacuolar degeneration in stressed adult cardiomyocytes, and in HF with reduced ejection fraction (HFrEF). All data points are shown as box and whisker plots (lines showing median, 25th and 75th percentiles with whiskers showing minimum and maximum values in: (**A**) phenylephrine stressed adult cardiomyocytes, * *p* < 0.05 vs. ACM-Control, and ^†^
*p* < 0.05 vs. ACM-PE, 2h; (**B**) rat model of pressure overload induced HFrEF, * *p* < 0.05 vs. Rat-Normal; and (**C**) human HFrEF, * *p* < 0.05 vs. Human-Normal. Adapted with permission from Chaanine et al. [11]. ACM: adult cardiac myocyte. PE: phenylephrine.

**Figure 4 jcm-09-03582-f004:**
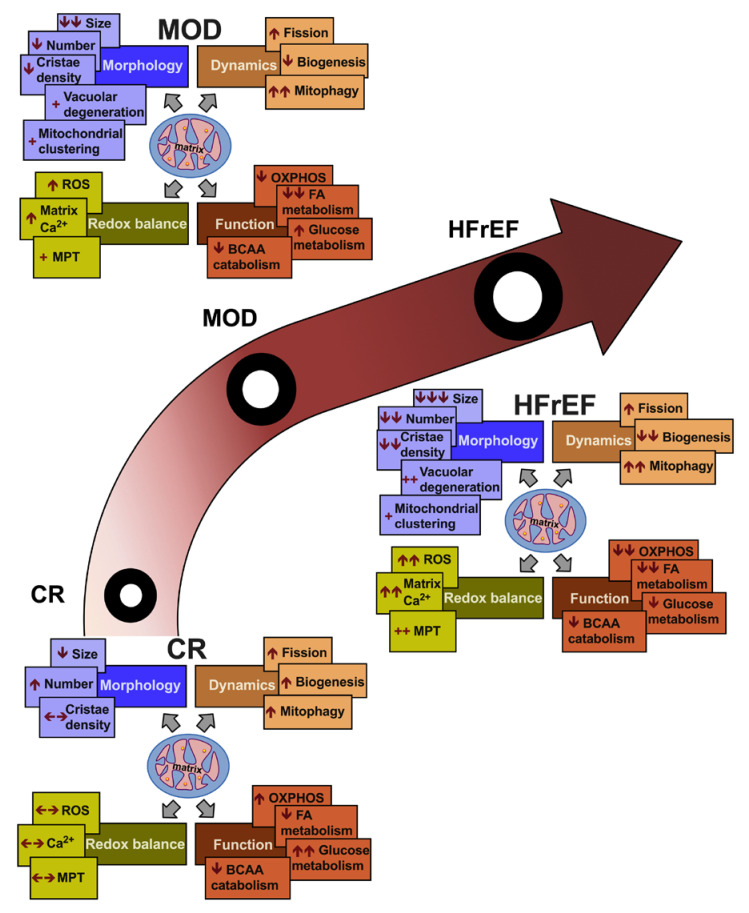
Central illustration of perturbations in mitochondrial physiology and metabolism in compensated hypertrophy and in progression to overt systolic heart failure. Mitochondrial fission and changes in substrate utilization and metabolism in favor of glucose are evident early on in pathological compensated hypertrophy (CR: concentric remodeling). Transitioning to a decompensated state, moderate remodeling and early systolic dysfunction (MOD HF phenotype), is associated with mitochondrial fragmentation, enhanced degradation and decreased oxidative capacity. Derangements in fatty acid (FA) beta-oxidation and pyruvate metabolism are evident at this stage of HF, along with increases in mitochondrial matrix calcium (Ca^2+^) and reactive oxygen species (ROS). Progression to advanced, overt, systolic HF (HFrEF) is associated with severe impairment in mitochondrial function and decrease in mitochondrial biogenesis and mass, FA and glucose metabolism. OXPHOS: oxidative phosphorylation, MPT: mitochondrial permeability transition, BCAA: branched chain amino acids.

**Figure 5 jcm-09-03582-f005:**
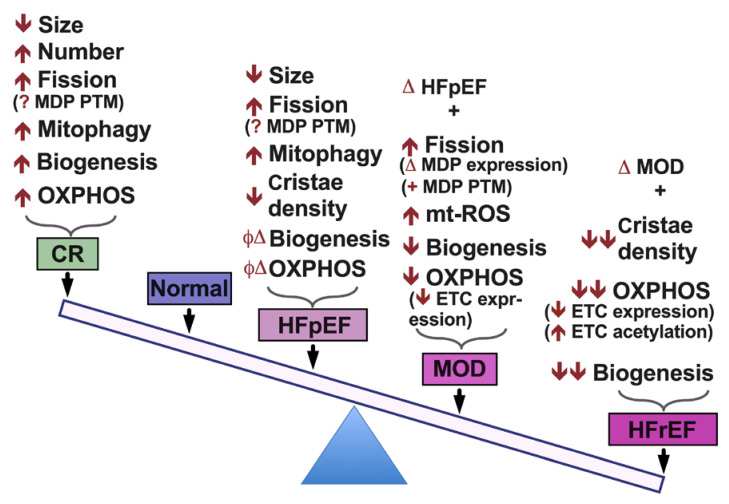
Illustration of key mitochondrial pathological findings in compensated hypertrophy and in progression to overt systolic heart failure. MDP: mitochondrial dynamic proteins, PTM: post-translational modification, ETC: electron transport chain Φ: No, Δ: Change.

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
