# Peer review of "Mitochondrial Pathobiology and Metabolic Remodeling in Progression to Overt Systolic Heart Failure"

_jcm, 2020, doi:10.3390/jcm9113582_

Round 1

Reviewer 1 Report

The review summarizes the role of mitochondria in heart failure, emphasizing its role in the development and progression of the disease. The field is of relevant interest because it can stimulate further research for novel therapeutic targets.

Comments:

  • There is confusion in the text about the use of abbreviations describing heart failure with reduced ejection fraction (sometimes HFrEF, sometimes SHF). I suggest using HFrEF because it’s more familiar in the clinical arena and endorsed by ESC guidelines.
  • It would be interesting to correlate the Stages of Mitochondrial vacuolar degeneration with AHA/ACC Stages of HF (from A to D)
  • Page 3, line 131: “These mitochondrial morphological changes were also observed in human HF and were more severe in patients with HF and reduced ejection fraction (HFrEF) compared to patients with HF and preserved ejection fraction (HFpEF) related to ischemic heart disease or severe aortic stenosis(9)”. I agree with this statement. Indeed, we demonstrated non-invasively the possibility of measuring the energetic efficiency of the cardiac pump work using the peak cardiac power output-to-mass, which is significantly reduced in HFrEF and proved to be an independent predictor of adverse outcome
  • In relation to the previous point, HFpEF is indeed characterized by different pathophysiology, and the role of the periphery seems to be relevant. Prior non-invasive (cite: 10.1093/eurjhf/hfr133; 10.1093/ehjci/jez014; 10.1002/ejhf.1739) and invasive (cite: 10.1161/CIRCHEARTFAILURE.114.001825) studies found that a peripheral limitation was the most important cause of reduced peak VO2 in HFpEF, whereas impaired cardiac output had less impact.
  • Please discuss the potential study of mitochondria in peripheral muscles as a novel and attractive therapy in this increasingly common class of HF (i.e. HFpEF), which at the moment is orphan of evidence-based therapy.
  • The central illustration should be improved, as the written part is cumbersome and difficult to read as a whole.

General comments:

  • English can be improved, starting from the abstract

Author Response

Reviewer 1:

There is confusion in the text about the use of abbreviations describing heart failure with reduced ejection fraction (sometimes HFrEF, sometimes SHF). I suggest using HFrEF because it’s more familiar in the clinical arena and endorsed by ESC guidelines.

We have modified the manuscript according to the reviewer’s suggestion. SHF was replaced with HFrEF, throughout the manuscript. Moreover, figure 3B was corrected to include Rat-HFrEF instead of Rat-SHF.

It would be interesting to correlate the Stages of Mitochondrial vacuolar degeneration with AHA/ACC Stages of HF (from A to D)

We have included the NYHA class of the patients from whom subepicardial biopsy (page 4, line 162-165). HFrEF patients/subjects were NYHA class III-IV, whereas, HFpEF patients were mainly NYHA class II or IIb.

Please discuss the potential study of mitochondria in peripheral muscles as a novel and attractive therapy in this increasingly common class of HF (i.e. HFpEF), which at the moment is orphan of evidence-based therapy.

We have included a paragraph highlighting this at the end of the manuscript before the Conclusion section, P;14, line: 404-412. We have referenced related literature, but did not expand significantly as it is out of the scope of the paper.

The central illustration should be improved, as the written part is cumbersome and difficult to read as a whole.

We have modified figure 4 and added a new figure, figure 5, highlighting key mitochondrial pathophysiological processes at each stage in transition to overt systolic HF.

English can be improved, starting from the abstract

We have edited certain paragraphs of the manuscript for more clarity including the Abstract.

Reviewer 2 Report

The authors in the present manuscript summarized the importance of altered (structural and functional) mitochondrial function during cardiac hypertrophy and different stages of heart failure. The present manuscript enhances the readers understanding on mitochondrial dysfunction in heart failure.

Here are my suggestions. 

1) At many places, the manuscript was not backed up by literature (citations). I would suggest authors refer them to the literature with appropriate citations

2) The authors centralized this manuscript only on cardiac hypertrophy and heart failure. Including pathophysiology of mitochondrial dysfunction in other cardiac diseases such as cardiomyopathy, arrhythmias, myocardial infarction etc.. will enhance the broader understanding of the mitochondrial dysfunction role in different cardiac diseases 

3) Graphical illustrations of key mechanisms in mitochondrial dysfunction during cardiac hypertrophy and heart failure would help. Figure 4 is good but only giving general idea 

Author Response

Reviewer 2:

At many places, the manuscript was not backed up by literature (citations). I would suggest authors refer them to the literature with appropriate citations

We have done so especially in the last section of the manuscript on “Metabolic remodeling in HF”.

The authors centralized this manuscript only on cardiac hypertrophy and heart failure. Including pathophysiology of mitochondrial dysfunction in other cardiac diseases such as cardiomyopathy, arrhythmias, myocardial infarction etc.. will enhance the broader understanding of the mitochondrial dysfunction role in different cardiac diseases

We have included a paragraph highlighting this at the end of the manuscript before the Conclusion section, P;14, line: 402-404. We have referenced related literature, but did not expand significantly as it is out of the scope of the paper.

Graphical illustrations of key mechanisms in mitochondrial dysfunction during cardiac hypertrophy and heart failure would help. Figure 4 is good but only giving general idea

We have modified figure 4 and added a new figure, figure 5, highlighting key mitochondrial pathophysiological processes at each stage in transition to overt systolic HF.

Round 2

Reviewer 1 Report

The authors responded to the previous comments.
I suggest to include the following refs (line 388):

10.1002/ejhf.1739
10.1161/CIRCHEARTFAILURE.114.001825